# Rethinking the Dilated Encoder in TE-YOLOF: An Approach Based on Attention Mechanism to Improve Performance for Blood Cell Detection

**DOI:** 10.3390/ijms232113355

**Published:** 2022-11-01

**Authors:** Fanxin Xu, He Lyu, Wei Xiang

**Affiliations:** Key Laboratory of Electronic and Information Engineering, State Ethnic Affairs Commission, Southwest Minzu University, Chengdu 610041, China

**Keywords:** blood cell detection, TE-YOLOF, attention mechanism

## Abstract

Blood cell detection is an essential branch of microscopic imaging for disease diagnosis. TE-YOLOF is an effective model for blood cell detection, and was recently found to have an outstanding trade-off between accuracy and model complexity. However, there is a lack of understanding of whether the dilated encoder in TE-YOLOF works well for blood cell detection. To address this issue, we perform a thorough experimental analysis and find the interesting fact that the dilated encoder is not necessary for TE-YOLOF to perform the blood cell detection task. For the purpose of increasing performance on blood cell detection, in this research, we use the attention mechanism to dominate the dilated encoder place in TE-YOLOF and find that the attention mechanism is effective to address this problem. Based upon these findings, we propose a novel approach, named Enhanced Channel Attention Module (ECAM), based on attention mechanism to achieve precision improvement with less growth on model complexity. Furthermore, we examine the proposed ECAM method compared with other tip-top attention mechanisms and find that the proposed attention method is more effective on blood cell detection task. We incorporate the spatial attention mechanism in CBAM with our ECAM to form a new module, which is named Enhanced-CBAM. We propose a new network named Enhanced Channel Attention Network (ENCANet) based upon Enhanced-CBAM to perform blood cell detection on BCCD dataset. This network can increase the accuracy to 90.3 AP while the parameter is only 6.5 M. Our ENCANet is also effective for conducting cross-domain blood cell detection experiments.

## 1. Introduction

### 1.1. Background

Blood cells are an important element in maintaining the normal functioning of human organs and tissues. Through the analysis of blood cells in microscopic images by identifying the cell types and distributions, doctors can make an accurate diagnosis of various diseases, such as blood disorders, polycythemia, and so on [1]. As for the auxiliary tools for the analysis of microscope images, slide scanners are used to generate super-resolution whole slide images (WSI) [2], which can realize the digitization of microscope images, thereby assisting physicians in disease diagnosis. However, this tool is to transfer the microscope image to the computer, where the doctor can judge the digitized image—it does not change the essence of manual statistics.

The application of machine learning in cell biology is the first attempt to solve the above artificial analysis problem [3]. Classical machine learning methods have been used to address blood cell detection problems, such as K-Nearest Neighbor (KNN), Naive Bayes (NB), Support Vector Machine (SVM), and so on [4,5,6]. The limitation with these methods is that they need to manually process image features first, then perform the blood cell detection task. Deep learning, developed from machine learning, is a representation learning method that can learn representations from raw data by multiple processing layers [7]. This method naturally addresses the limitations of machine learning methods and elevates blood cell detection to the degree that it can be deployed in an end-to-end way. With the power of deep learning, doctors can avoid the manual positioning of image information and focus more on information analysis. For researchers, they can allocate more attention to addressing the problem of adaptability and accuracy of deep network models in blood cell detection. The initial methodology for researchers is to directly use the original deep network model to process and analyze the blood cell detection dataset, using either object detection methods [8,9] or semantic segmentation methods [10,11].

In this paper, we focus more on the application of object detection methods in blood cell detection. Kutlu et al. proposed a convolutional neural network based on R-CNN and transfer learning methods to achieve blood cell detection [8]. Zhang et al. used the YOLOv3 detector for blood cell detection [9]. The direct transfer of original models to new domains for detection is effective, but they are relatively expensive on model complexity. Recently, many lightweight models have appeared in the field of medical detection analysis [12,13], as well as in the blood cell detection task [14,15]. Our research focus in this paper is on the recently proposed model TE-YOLOF [15], which is an effective model for blood cell detection with an outstanding trade-off between accuracy and model complexity. In the process of understanding the architecture of the model, a question naturally arises: is the dilated encoder a necessary component in TE-YOLOF for blood cell detection task? See Figure 1 for a detailed comparison under the premise of TE-YOLOF with or without the dilated encoder module. Figure 1 clearly shows that the dilated encoder does not contribute much to the performance improvement of blood cell detection but occupies a part of the limited model complexity.

Based on this finding, we directly remove the dilated encoder in TE-YOLOF. At this point, another interesting question naturally arises: can we improve the accuracy of the model while not increasing the model complexity greatly? Our intention is to improve the performance without affecting the overall architecture of the network; then, this question evolves into whether it is possible to design a module and place it in the position where the dilated encoder is located so as to achieve the effect of improved performance.

The above question motivates us to pay attention to the attention mechanism. SENet [16] and CBAM [17] are the representative networks that use the channel attention mechanism and spatial attention mechanism, respectively. These attention mechanism approaches have been successful used in other medical analysis problems [18,19]. See Section 1.2.2 for details on the application of the attention mechanism. We attempt to incorporate these attention mechanisms into TE-YOLOF. Later experiments can prove that integrating them in our network can effectively enhance the detection performance. Based upon these findings, we deeply analyze how these mechanisms are effective and propose a new approach based on the channel attention mechanism, which is named Enhanced Channel Attention Module (ECAM). See Section 4.1 for specific module details of ECAM and the source of improvement idea. We examine the proposed ECAM method compared with other attention mechanisms to prove that the proposed attention method is effective on addressing blood cell detection problem. In a nutshell, the contributions of this paper can be summed up as the following items:Through experiments, we prove that the dilated encoder is a non-essential component in TE-YOLOF to address the problem of blood cell detection. This finding can reduce nearly one-third of the parameters of the original model, which reduce the model complexity from 9.94 M to 6.46 M.Attention mechanism is an effective way to increase model TE-YOLOF detection performance without greatly increasing model complexity on the blood cell detection task.The proposed attention mechanism ECAM can perform better on blood cell detection than other attention methods, such as SENet, CBAM, and so on. TE-YOLOF-B0 is still only 6.49 M after incorporating this module. Even the model complexity of ECBAM, which adds the spatial attention mechanism in CBAM into ECAM, is only 6.5 M.

### 1.2. Related Work

#### 1.2.1. Blood Cell Detection

Blood cell detection can provide necessary information for the diagnosis of various diseases. With the advent of the era of big data and the digitization of blood cell pictures through WSI, a large number of microscope images can be generated, which provides fertile soil for the growth of automatic data analysis methods [20]. A typical blood cell tissue has three components: white blood cells (WBCs), red blood cells (RBCs), and platelets [21]. For their analysis, the first attempt is to use machine learning methods to automate the blood cell analysis, so as to achieve computer-aided goals. The initial blood cell detection system has mainly three steps: (1) Detection of blood cells from the image. (2) Extracting features from the detection area. (3) Designing a classifier to perform the blood cell classification task [22]. Kumar et al. used the K-Nearest Neighbor and Naive Bayes Classifier to perform the blood cell detection task [4]. Markiewicz et al. proposed a classifier based on Support Vector Machine to conduct blood cell classification, which is the innovation of their third step [6]. Mandal et al. proposed a classifier based on Gradient Boosting Decision Tree (GBDT) for blood cell detection [23].

The object detection method based on deep learning takes the above machine learning methods a big step forward, combining the three steps of the traditional blood cell detection system into one step. The detection system only needs to perform detection on the original image to obtain the desired effect. Because of this simplicity of thought, researchers have tried to transfer the two-stage and one-stage object detection networks for addressing blood cell detection problems. Kutlu et al. proposed a convolutional neural network based on regional convolutional neural network, which is the essential component of the two-stage object detection network, to achieve blood cell detection [8]. Zhang et al. used the YOLOv3 detector, which is the representative one-stage object detection network, for blood cell detection [9]. The difference in medical problems compared with general object detection problems is that they require a network with lower model complexity and good detection effect to meet a wider range of usage scenarios, as well as in blood cell detection problems. For the purpose of meeting this demand, Shakarami et al. made a lightweight improvement to YOLOv3 for blood cell detection [14]. On the basis of the above improvements, Xu et al. proposed a lightweight TE-YOLOF to further improve the accuracy and reduce the model complexity to less than 10 M [15]. The research in this paper is to further reduce the model complexity on the basis of TE-YOLOF.

#### 1.2.2. Attention Mechanisms

The application of attention mechanisms in the field of computer vision mainly involves channel attention mechanism and spatial attention mechanism. SENet [16] and CBAM [17] are the representative modules for the above two attention mechanisms separately. They are initially used to improve the accuracy of object classification task by enhancing the expressiveness of feature maps. These attention mechanism modules and their variants can be applied to many other tasks, not limited to object classification tasks. Li et al. proposed FCAM that adds one Multilayer Perceptron (MLP) in channel attention layer to improve performance on point cloud detection [24]. Hu et al. proposed a Twin attention mechanism that updates the channel attention and the spatial attention in CBAM to parallel and uses this method to carry out surface defect classification [25]. Zhu et al. used the SE block to perform enhancing feature task in region channel [26].

The application of attention mechanism is not limited to the above questions, they also have been used in medical analysis problems as well as in blood cell detection problems. Tong et al. proposed ASCU-Net, which places the attention gate before CBAM to address the skin lesion problem [18]. Jiang et al. integrated CBAM into each stage output in YOLOv3 [27] for feature enhancement to perform blood cell detection [28]. Our research focuses on integrating attention mechanism into TE-YOLOF to improve the performance of blood cell detection tasks.

## 2. Results

### 2.1. Implementation Details

In this work, the implementation of our network is the same as TE-YOLOF in [15]. The overall architecture is implemented by Pytorch and uses an Nvidia GeForce 1080Ti to train our network. The training strategy we use is SGD and the mini-batch setting is 4. All models are trained under an initial learning rate of 0.12 with 12 epochs. We set the warmup iterations as 1500 at the beginning to stabilize the training. In the post-processing, we use the threshold of 0.6 of NMS to process results. For the evaluation metrics, we use the MS COCO detection evaluation metrics in all our experiments.

### 2.2. Dataset

The BCCD dataset https://public.roboflow.com/object-detection/bccd (accessed on 26 July 2021), which includes 364 images, is used to conduct the blood cell detection task, which contains three cell objects: platelets, white blood cells (WBC), and red blood cells (RBC). All the images in this dataset are 416×416×3, after being resized from the original dimensions of 640×480×3. The division protocol of this dataset divided images into training, val, and test sets at a ratio of 7:2:1. In addition, data augmentation strategies such as horizontal flip and vertical flip are used to expand the diversity of images. In order to prove that our network can achieve cross-domain blood cell detection, the BCDD dataset https://github.com/draaslan/blood-cell-detection-dataset (accessed on 16 October 2021) and the WBC Image Dataset (WID) https://github.com/zxaoyou/segmentation_WBC (accessed on 18 October 2021) are used for generalization verification. All datasets are annotated in COCO format.

### 2.3. Comparison with TE-YOLOF

For fair comparison with TE-YOLOF, we use the same settings for TE-YOLOF as in [15]. The only difference in this comparison is that the dilated encoder is changed to the proposed method and other attention methods. In the ablation experiments below, we can find that the spatial attention mechanism is useful for improving the performance of blood cell detection; so, we use the proposed ECBAM, which is ECAM with the spatial attention mechanism, to perform the comparison task. In each version of the backbone in TE-YOLOF, we also conduct the comparison. The comparison results can be seen in Table 1.

According to the results in Table 1, it is obvious that TE-YOLOF can achieve blood cell detection well under different versions of backbone without dilated encoder, but this change can bring a litter reduced accuracy in $AP_{50}$. Without dilated encoder, TE-YOLOF-B0 can reduce the parameters from 9.94 M to 6.46 M (−3.48 M) and the GFLOPS from 6.21 G to 2.73 G (−3.48 G), the other versions from EfficientNet-B1 to EfficientNet-B3 also reduced the parameters at 3.48 M and the GFLOPS at 3.48 G. Then, in each version of backbone, we use the TE-YOLOF without dilated encoder as baseline. With the power of the spatial attention mechanism in CBAM, the accuracy of $AP_{50}$ can improve 0.1–0.4 AP based upon the different backbone. Meanwhile, when we introduce the channel attention mechanism, the accuracy of $AP_{50}$ can be improved by 0.2–0.6 AP based upon the different backbone. All versions of modified TE-YOLOF can improve the accuracy of $AP_{50}$ higher than 90.0; further, they can all be stable in the range of 90.0 to 90.5. Such blood cell detection results are already comparable with the detection results of the original TE-YOLOF. The detection result of TE-YOLOF-B0 is 90.3 AP, which is lower than the detection result 90.5 AP of TE-YOLOF-B3 by only 0.2 AP. Comparing the 1.2 AP gap of the original TE-YOLOF between the EfficientNet-B0 and EfficientNet-B3 backbones, this comparison result shows that the proposed method ECBAM can give the power of stable detection performance for different backbones. From this comparison, it can be clearly seen that the parameters can be reduced from 13.31 M to 6.5 M and the GFLOPS can be reduced from 3.12 G to 2.73 G. From the perspective of computational complexity, this method saves a lot of computational resources. ECBAM has the same power of improving the accuracy on $AP_{50}$ compared with CBAM, and it can increase on $AP_{75}$ by 1.8 to 6.5 AP compared with CBAM.

### 2.4. Ablation Experiments

In this section, we run a number of ablations to analyze the rationality of the proposed ECAM and ECBAM. We first run the ablation of different channel attention modules to analyze whether the proposed method of ECAM is more effective than other tip-top attention mechanisms. Then, we analyze the necessity of adding a spatial attention mechanism so as to form our proposed method of ECBAM. Finally, we perform spatial attention module kernel size ablation experiments to select the best kernel size for our ECBAM and perform the reduction ratio of MLP in ECAM ablation experiments to select the best reduction ratio in ECAM. For the credibility, all the ablation results are the average of three trials. The base model is TE-YOLOF-B0.

#### 2.4.1. Different Channel Attention Module

In the selection of the channel attention mechanisms, based upon the selection of SE and CAM in CBAM, we also add an ECA [29] module, which is improved based on SE, to perform the ablation study. A total of three attention mechanisms are selected. Details can be seen in Table 2. The ECA module is not effective for improving performance of the blood cell detection task, even if it does not increase the parameters and the GFLOPS. The SE module and the CAM in CBAM both can improve the performance by 0.2–0.3 AP. Meanwhile, our proposed method of ECAM module can improve the accuracy to 90.1 under the premise of the same model complexity with the above two attention mechanisms. With the power of the Enhanced Channel Attention Mechanism, $AP_{50}$ can be increased from 89.7 to 90.1, while the parameters are increased by 0.3 M and GFLOPS stay the same. So, the proposed method of ECAM is the most effective attention mechanism in blood cell detection task.

#### 2.4.2. Add Spatial Attention Module or Not

The spatial attention mechanism we used is the spatial attention mechanism in CBAM. For the validity of the ablation experiments for the spatial attention mechanism, we add the spatial attention mechanism to the above effective channel attention mechanism to validate whether to add the spatial attention module or not. We select the SE block, CAM in CBAM, and the proposed ECAM as the base channel attention mechanism. We add the spatial attention mechanism after these channel attention mechanisms to perform the ablation study. The results are shown in Table 3. The spatial attention mechanism can increase the parameters by 0.01 M while the GFLOPS is stable with 2.73 G. With the power of the spatial attention mechanism, each module can improve the accuracy by 0.2–0.4 AP. Further, our proposed ECBAM can improve the accuracy from 89.7 AP to 90.3 AP, which is the highest AP.

#### 2.4.3. Spatial Attention Module Kernel Size

The ablation study of spatial attention module kernel size is used to select the convolution kernel size of the spatial attention mechanism that is most conducive to blood cell detection. The results are shown in Table 4. It is obvious that the accuracy increases linearly as the kernel size increases from 3 to 7, while it decreases linearly from 7 to 11. The kernel size in the range of 5 to 9 can lead to performance improvement in blood cell detection. Further information we can find is that when the kernel size is too large or too small, it will have a negative effect on the improvement of accuracy. The best effect can be achieved at 90.3 AP when the kernel size is 7. Therefore, 7 is the best kernel size with the best receptive field for the spatial attention mechanism to extract spatial information for blood cell detection.

#### 2.4.4. Reduction Ratio of MLP in ECAM

The base model is the ENCANet with the ECBAM, and the ablation study is on the reduction ratio of MLP in ECAM, which is the channel attention mechanism in ECBAM. The ablation experiments are shown in Table 5. According to Table 5, the reduction ratio of MLP in the range of 8 to 32 can give the power for ECBAM to improve the performance. The reduction ratio in a small number could have a negative effect on ECBAM to perform blood cell detection. It is obvious that the best reduction ratio for ECAM is 16, which helps the ENCANet to achieve the accuracy of 90.3 AP, while the parameters are only 6.5 M. When the reduction ratio of MLP is 16, it can maintain the model complexity without drastic changes on the basis of extracting channel attention information to the maximum extent.

### 2.5. Visual Results

For the qualitative analysis, visual results of different attention mechanisms are shown in Figure 2 and the quantitative results are given in Table 2 and Table 3. The base model we used is the proposed ENCANet with EfficientNet-B0 as backbone, which has the minimum model complexity. The attention maps in Figure 2 are calculated for the outputs of the used model with the introduction on the left. For the analysis, we aggregate the attention maps of all modules together for comparison. In Figure 2, it can be clearly seen that the ECAM-integrated network and ECBAM-integrated network cover the blood cell object regions better than other methods—that is, the ECAM-integrated network and ECBAM-integrated network can perform the feature enhancement task more effectively and comprehensively than other attention mechanisms.

Based on the above analysis, we use the ECBAM-integrated network, which is named ENCANet, to perform the blood cell detection task on BCCD test dataset and other blood cell datasets such as BCDD dataset and WID dataset. The detection results on BCCD test dataset of the proposed ENCANet can be seen in Figure 3. The BCDD dataset and WID dataset are used to perform generalization verification of our proposed network for other blood cell detection tasks. The visualization detection results can be seen in Figure 4 and Figure 5 separately. Through Figure 3, Figure 4 and Figure 5, it can be clearly observed that our proposed model ENCANet trained in BCCD dataset can perform the blood cell detection task on the BCCD test dataset well. Our ENCANet can be directly transferred for inference on other cross-domain blood cell datasets with excellent performance.

## 3. Discussion

When the slide scanners are used to perform digitization of microscope images, and Flow Cytometry Instrumentation [30] is used to perform an automated count task for blood cells traditionally, these methods are used to assist the physicians to conduct accelerated medical analysis. When the machine learning methods and the deep learning methods receive the microscope images of blood cells, these methods can perform blood cell detection without manual testing compared with traditional methods. Based on this purely automated method that has been separated from manual operations, it requires relatively high accuracy and low model complexity that can be used to perform blood cell detection in more complex environments.

In this research, we presented the lightweight model ENCANet to achieve 90.3 AP on blood cell detection with only 6.5 M parameters. In this new network, we incorporated the proposed ECBAM to perform the feature enhancement task; this new attention mechanism can improve the ENCANet by 0.6 AP with the price of only 0.04 M parameters. This new channel attention mechanism of ECAM in ECBAM can perform better than other tip-top channel attention mechanisms on blood cell detection; detailed results can be seen in Table 2. Attention mechanisms have been used to carry out medical image analysis in recent years [18,31]. The proposed method ECBAM retains the possibility of being applicable to other medical issues.

Our proposed model is used for detection on the raw images of blood cells, while recent researches have partly turned their attention to image quality that can be used to enhance the image content and preserve information. By using medical image fusion technique to create a single image by combing multiple image modalities, these methods can be used as the potentially effective way to enhance original image contrast and object demarcation, such as EOA [32], MPA [11], and so on.

## 4. Materials and Methods

### 4.1. Enhanced Channel Attention Module

In this section, we first conduct a detailed analysis of the architecture of the Squeeze-and-Excitation (SE) Block in [16]. Then, we present a detailed comparison and analysis of the channel attention mechanism used in CBAM [17]. Finally, we introduce the design of our Enhanced Channel Attention Module (ECAM).

#### 4.1.1. Review of SENet

Squeeze-and-Excitation block is a powerful lightweight attention mechanism that focuses on enhancing the representation of the network by modeling channelwise relationships in a computationally efficient manner. This module is an efficient way to map input features $X\in R^{H\times W\times C}$ to output features $O\in R^{H\times W\times C}$ that embed the channel attention information. The transformation from *X* to *O* can be seen as the learned set of filter kernels, using $V=[v_{1},v_{2},...,v_{c}]$ to denote these filters, while $v_{c}$ refers to the parameters of the *c*-th filter. Then, we can write the output feature maps as $O=[o_{1},o_{2},...,o_{c}]$, where $o_{c}=v_{c}*x_{c}$. Here, $v_{c}$ is generally a scalar, and $x_{c}$ is an input feature map where $x_{c}\in R^{H\times W}$. The ∗ denotes scalar multiplication.

The essential goal of SE block in SENet is to generate the $V\in R^{1\times 1\times C}$, which is named the Channel Attention $P_{C}$ in Figure 6. Figure 6 is the diagram of the SE block to show how the channel attention is generated in visualization. The overall architecture of SE can be divided into two steps, *Squeeze* and *Excitation*, separately. The *Squeeze* step is to squeeze the global information into a channel descriptor by global average pooling. The channel descriptors $Z\in R^{C}$ are generated by global average pooling method from shrinking spatial dimensions H×W to 1×1. The *c*-th descriptor of Z can be calculated by
(1)$$z_{c}=F_{GAV}\left(x_{c}\right)=\frac{1}{H\times W}\sum_{i=1}^{H}\sum_{j=1}^{W}x_{c}\left(i,j\right)$$
where GAV denotes global average pooling. After squeeze, the descriptors $Z\in R^{1\times 1\times C}$ are the representative scalar for each channel of input feature maps. The *Excitation* step is to capture channelwise dependencies. This objective is implemented by two fully-connected layers with the non-linearity function. The detailed formula is as follows:(2)$$P_{C}=F_{Excitation}\left(Z,W\right)=\sigma \left(W_{2}\delta \left(W_{1}Z\right)\right)$$
where δ refers to the ReLU [33] function, σ refers to the Sigmoid function, $W_{1}\in R^{\frac{C}{r}\times C}$, and $W_{2}\in R^{C\times \frac{C}{r}}$. *r* is used as a reduction ratio for limiting model complexity. After channel feature aggregation by the *Excitation* step, the output channel descriptors could be in the range of $p_{c}\in \left(0,1\right)$. SE block is effective to achieve a performance improvement in TE-YOLOF for blood cell detection; the experiment is shown in Section 2.4.1.

#### 4.1.2. Motivated by CBAM

Inspired by the channel attention mechanism in SE block that can improve the performance of TE-YOLOF in the blood cell detection task, we turn our attention to the representative attention mechanism named CBAM that combines channel attention and spatial attention. This module is effective for improving blood cell detection performance while defeating the SE block; the detailed experiment can be seen in Table 3.

While CBAM is the attention mechanism that combines channel attention mechanism and spatial attention mechanism, a question that naturally arises is whether the channel attention in CBAM is effective for performance improvement in blood cell detection. Thus, we perform an experiment using CBAM module without spatial attention mechanism; we named this module Channel Attention Mechanism (CAM). The diagram of the CAM can be seen in Figure 7. According to the experiments result, CAM is also effective to perform the blood cell detection task.

The difference between SE block and CAM is that CAM uses the global max pooling method to perform the same feature aggregation as the global average pooling method additionally. We define the channel descriptors of global average pooling as the above SE block used, $Z\in R^{C}$. We define the channel descriptors of global max pooling as $M\in R^{C}$. The *c*-th descriptor of *M* can be calculated as follows:(3)$$m_{c}=F_{MAX}\left(x_{c}\right)=\max_{\left(i,j\right)\in R^{H\times W}}x_{c}\left(i,j\right)$$ The two channel descriptors of *Z* and *M* are applied to perform feature aggregation by the shared multilayer perceptron (MLP) with one hidden layer; then, they can be transformed to two different channel context descriptors: $F_{avg}\left(X_{C}\right)$ and $F_{max}\left(X_{C}\right)$. In short, the channel attention mechanism can be computed as below:(4)$$\begin{array}{cc} P_{C} & =\sigma (MLP(MaxPool(X))+MLP(AvgPool(X)))\\ \; & =\sigma (W_{2}\delta \left(W_{1}\left(F_{avg}\left(X_{C}\right)\right)+W_{2}\delta \left(W_{1}\left(F_{max}\left(X_{C}\right)\right)\right)\right) \end{array}$$
where σ denotes the sigmoid function, $W_{1}\in R^{\frac{C}{r}\times C}$ and $W_{2}\in R^{C\times \frac{C}{r}}$, and δ is the ReLU activation function. According to the difference between Equations (Equation 2) and (Equation 4), with the visual diagram of attention mechanisms in Figure 6 and Figure 7, it is obvious that the global max pooling method is one effective trick to enhance feature aggregation. In addition to using the global max pooling method like the usage of global average pooling method, what about using the global max pooling method to enhance the features obtained by global average pooling method? This motivated us to design a new channel attention mechanism named Enhanced Channel Attention Mechanism (ECAM), see the section below for details.

#### 4.1.3. Enhanced Channel Attention Mechanism

By comparing the difference between SE block and the CAM in CBAM, it is obvious that global max pooling is an effective branch to enhance the final channel attention. Since it can enhance the final channel attention, can we use it to enhance the global average pooling branch, so as to achieve the effect of reducing an MLP calculation and improving performance? This idea motivated us to design our Enhanced Channel Attention Module (ECAM) below. On the basis of ECAM and the spatial attention mechanism used in CBAM, we also designed the ECBAM, which contains the channel attention mechanism and the spatial attention mechanism to further improve performance on blood cell detection.

##### Enhanced Channel Attention Module

The essence of Enhanced Channel Attention Module is using the global max pooling method with sigmoid function to enhance channel descriptors generated by global average pooling method. The detailed diagram of our proposed channel attention module can be seen in Figure 8. The channel attention descriptors $Z^{\prime }\in R^{C}$ generated can be seen as the formula below to enhance the *Squeeze* step in SE block. The *c*-th descriptor of $Z^{\prime }$ can be calculated as below:(5)$$z_{c}^{\prime }=F_{ECA}\left(F_{GAV}\left(x_{c}\right)\right)=\sigma \left(F_{MAX}\left(x_{c}\right)\right)*F_{GAV}\left(x_{c}\right)$$
where $F_{MAX}$ is the method used in Equation (Equation 3) and $F_{GAV}$ is the method used in Equation (Equation 1). Then, the *Excitation* step can be calculated as below:(6)$$P_{C}=F_{Excitation}\left(Z^{\prime },W\right)=\sigma \left(W_{2}\delta \left(W_{1}Z^{\prime }\right)\right)$$
where σ denotes the sigmoid function, $W_{1}\in R^{\frac{C}{r}\times C}$ and $W_{2}\in R^{C\times \frac{C}{r}}$, and δ is the ReLU activation function.

##### Enhanced CBAM

Motivated by CBAM and the fact that the channel attention can be serially combined with the spatial attention mechanism, we adopt the same architecture as CBAM and replace the channel attention mechanism in CBAM with our Enhanced Channel Attention Mechanism so as to form the new Enhanced-CBAM attention mechanism. The detailed spatial attention mechanism used in CBAM can be seen in Figure 9. We define the feature maps refined by the channel attention mechanism as $T=[t_{1},t_{2},...,t_{c}]$, which is the abbreviation of temporary feature maps. We describe the detailed operation of the spatial attention below.

Unlike the aggregation method of the channel attention mechanism in spatial dimension, the spatial attention aggregates channel information by using two pooling methods, which are the average pooling and the max pooling, to generate two 2D feature maps: $F_{avg}^{S}\in R^{H\times W\times 1}$ and $F_{max}^{S}\in R^{H\times W\times 1}$. These two feature maps can be concatenated and converged by a standard convolution layer to produce a 2D spatial attention map $P_{S}\in R^{H\times W\times 1}$. The spatial attention can be calculated as below:(7)$$P_{S}=\sigma \left(f^{7\times 7}\left(\left[F_{avg}^{S}\left(T\right);F_{max}^{S}\left(T\right)\right]\right)\right)$$
where σ denotes the sigmoid function and $f^{7\times 7}$ represents the standard convolution with the filter size of 7×7.

The above context is the introduction of the method of spatial attention mechanism we used; then, we give a brief introduction to the channel attention mechanism and spatial attention mechanism in a sequential manner, which we named Enhanced-CBAM (ECBAM). The overall architecture of ECBAM can be seen in Figure 10.

### 4.2. Enhanced Channel Attention Network

Based upon the proposed attention mechanism method and the TE-YOLOF used in [15], we divided the architecture of our network into three components: Backbone, Encoder, Decoder. The detailed overall architecture can be seen in Figure 11. We assume that the input is N×H×W×C, where *N* represents the batch of input images, *H* and *W* represent the height and weight of input image, and *C* represents the channels. We attached the input and output size after the introduction of each component.

#### 4.2.1. Backbone

Backbone is used to perform feature extraction. The choice of the network used in backbone is generally the network pretrained on a large dataset. In our network, we use the same backbone used in [15]: EfficientNet [34]. The default is EfficientNet-B0; EfficientNet-B1 to B3 also can be used to perform feature extraction task in our network. This ensures flexibility in the selection of network. The input images could be a fixed size of N×416×416×3.

#### 4.2.2. Encoder

Encoder is used to perform feature enhancement. The component we designed for it is the proposed method above. The Enhanced Channel Attention Module and Enhanced CBAM can be placed in the Encoder location. In Figure 11, we use the ECBAM as default. The input size is the fixed channel number of N×512×H×W. These *H* and *W* represent the height and weight of feature map from backbone. The output size of this component is also N×512×H×W.

#### 4.2.3. Decoder

Decoder is used to perform potential object localization. The decoder we used is the same as the decoder used in [15], which consists of two parallel heads to perform classification and regression separately. The two heads consist of different numbers of Depthwise Separable Convolution Modules that combine the depthwise separable convolution and residual connection. The details of Depthwise Separable Convolution Module that represent the DW Module in Figure 11 can be seen in [15]. In decoder, *K* represents *K* object categories and *A* stands for *A* anchors in each pixel of the feature map.

## 5. Conclusions

In this research, we presented the Enhanced Channel Attention Module (ECAM), a novel approach based on attention mechanism, to improve performance on blood cell detection. The approach achieves comparable results to other attention mechanisms on the BCCD dataset. We incorporated the spatial attention mechanism in CBAM with ECAM to form Enhanced-CBAM and used it to construct a new network named Enhanced Channel Attention Network (ENCANet). The results of detailed experiments and comparisons demonstrate that ENCANet is more effective than other existing methods for blood cell detection with only 6.5 M parameters. ENCANet can also be generalized to perform the detection task for other blood cell datasets. However, our model may not be effective in more complex scenarios such as overcrowded blood cells in image. We hope our ENCANet can perform blood cell detection with a variety of test scenarios and that ECAM can address other medical image problems in future research.

## Figures and Tables

**Figure 1 ijms-23-13355-f001:**
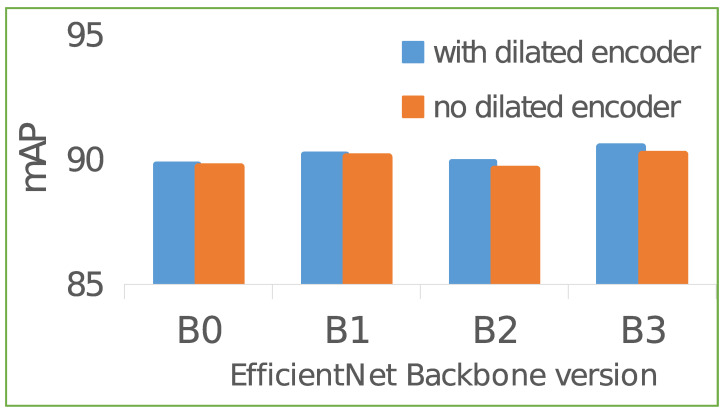
‘BN’ denotes that the backbone network used is EfficientNet-BN. We perform a thorough experimental analysis on the TE-YOLOF examining whether the dilated encoder can help in the blood cell detection on the blood cell detection BCCD dataset. This figure shows the performance comparison of the model on mAP with or without the dilated encoder module when replacing different backbones of TE-YOLOF. The metric of mAP represents that the mean Average Precision of each category with the Intersection Over Union (IoU) threshold of 0.5.

**Figure 2 ijms-23-13355-f002:**
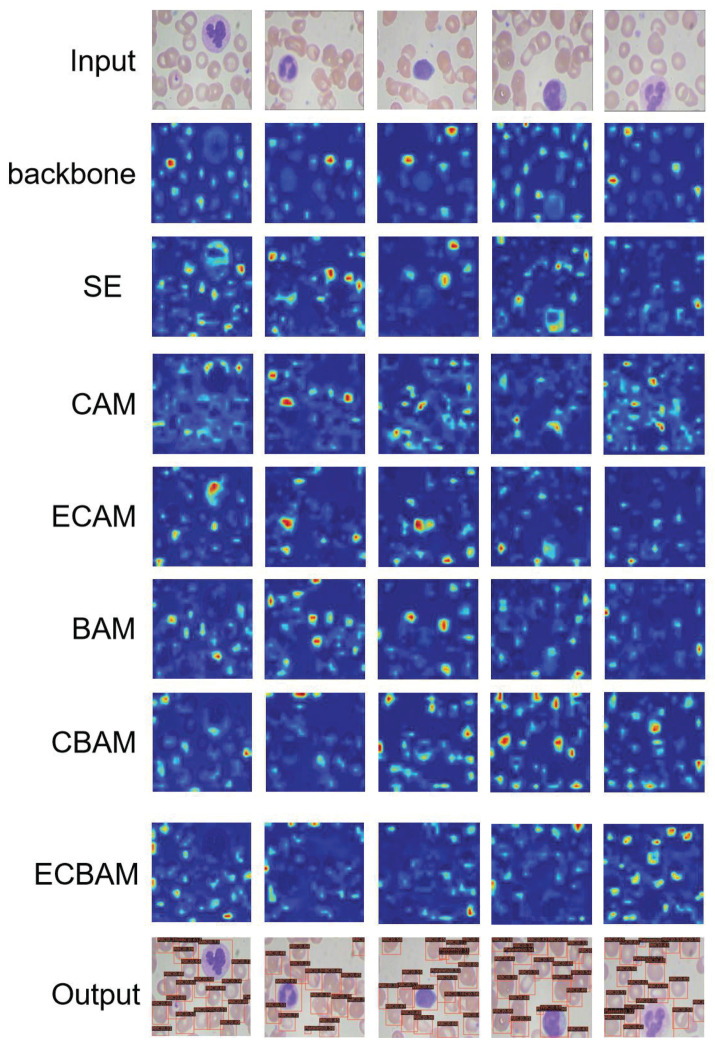
Attention map results on BCCD test set. Top to bottom: input images; the attention maps generated after backbone; the visualization results after the selected attention mechanisms, such as SE, and CAM in CBAM, ECAM, BAM, CBAM, ECBAM, and so on; the detection results of input images. The visualization is calculated after the selected module.

**Figure 3 ijms-23-13355-f003:**
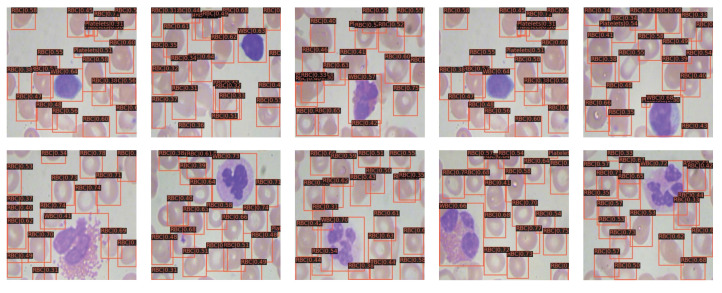
Visualization detection results on BCCD test dataset.

**Figure 4 ijms-23-13355-f004:**
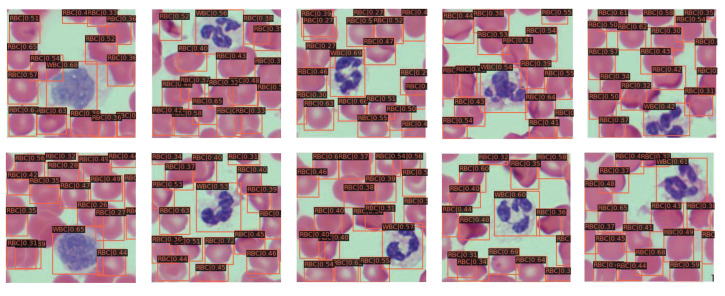
Visualization detection results on BCDD dataset.

**Figure 5 ijms-23-13355-f005:**
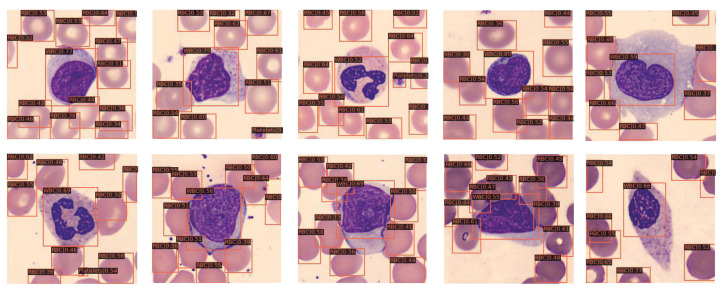
Visualization detection results on WID dataset.

**Figure 6 ijms-23-13355-f006:**
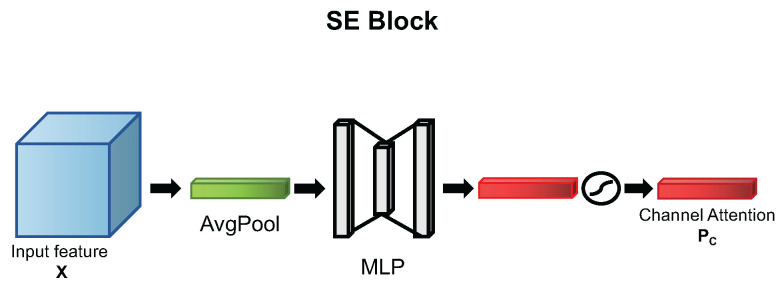
The channel attention mechanism used in SENet [16].

**Figure 7 ijms-23-13355-f007:**
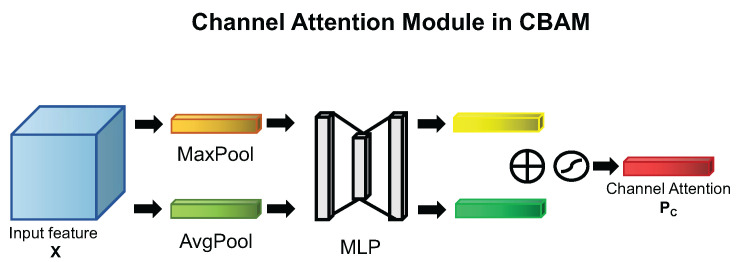
The channel attention mechanism used in CBAM [17].

**Figure 8 ijms-23-13355-f008:**
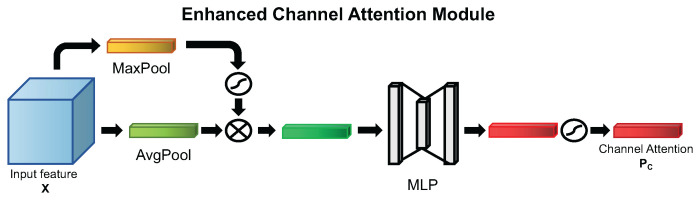
The proposed channel attention mechanism of ECAM.

**Figure 9 ijms-23-13355-f009:**
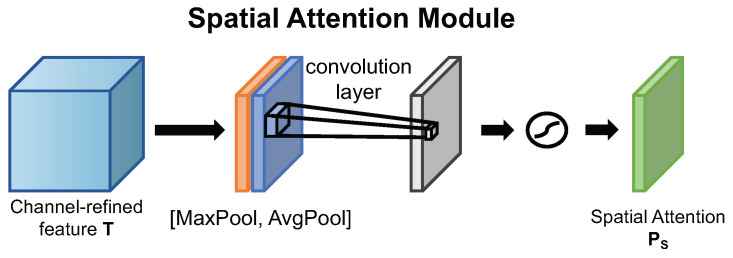
Diagram of spatial attention mechanism used in CBAM.

**Figure 10 ijms-23-13355-f010:**
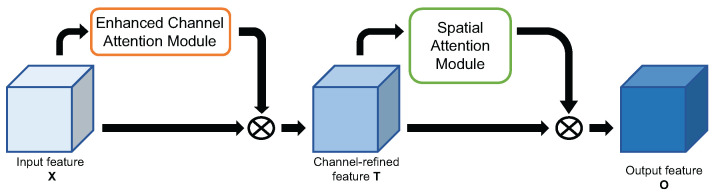
Diagram of Enhanced-CBAM.

**Figure 11 ijms-23-13355-f011:**
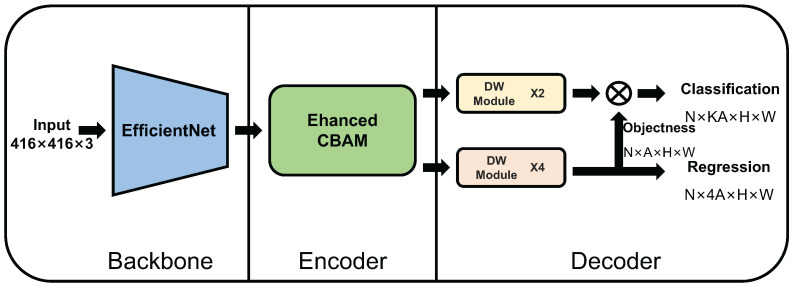
Overall architecture of the Enhanced Channel Attention Network.

**Table 1 ijms-23-13355-t001:** Comparison with TE-YOLOF on the BCCD validation set. Models marked with the suffix ‘B(N)’ adopt EfficientNet-B(N) as backbone by default. ‘- dilated encoder’ represents that TE-YOLOF does not use the dilated encoder to achieve feature enhancement, and uses the 1×1 conv to reduce the channels from backbone to the default number of 512. ‘+ BAM’ represents using the spatial attention mechanism in CBAM to achieve feature enhancement, while this module is placed in the original dilated encoder location. ‘+ CBAM’ represents using the original attention mechanism in [17], which contains the channel attention mechanism and the spatial attention mechanism. ‘+ ECBAM’ represents using the Enhanced Channel Attention Module we proposed with the spatial attention mechanism in CBAM.

Model	#Params	GFLOPS	AP	${\mathrm{AP}}_{50}$	${\mathrm{AP}}_{75}$	${\mathrm{AP}}_{S}$	${\mathrm{AP}}_{M}$	${\mathrm{AP}}_{L}$
TE-YOLOF-B0	9.94 M	6.21 G	55.6	89.4	59.8	35.1	58.2	48.1
- dilated encoder	6.46 M	2.73 G	56.1	89.7	59.5	37.1	59.9	47.3
+ BAM	6.46 M	2.73 G	55.5	89.8	57.2	36.1	58.8	44.1
+ CBAM	6.5 M	2.73 G	56.8	90.3	60.6	36.0	59.8	49.4
+ ECBAM	6.5 M	2.73 G	57.1	90.3	63.8	37.1	58.9	48.2
TE-YOLOF-B1	12.45 M	6.32 G	57.8	90.1	64.5	38.4	62.2	46.1
- dilated encoder	8.97 M	2.84 G	57.3	89.4	67.0	43.6	58.5	47.3
+ BAM	8.97 M	2.84 G	55.6	89.8	61.3	37.3	59.1	47.0
+ CBAM	9 M	2.84 G	55.5	90.4	59.5	39.6	58.9	44.1
+ ECBAM	9 M	2.84 G	57.2	90.4	66.0	39.3	60.0	44.6
TE-YOLOF-B2	13.7 M	6.4 G	56.6	90.1	60.4	37.1	59.1	47.8
- dilated encoder	10.22 M	2.92 G	57.0	89.9	62.0	37.6	59.1	46.9
+ BAM	10.22 M	2.92 G	58.3	89.9	62.0	40.3	60.2	46.8
+ CBAM	10.25 M	2.92 G	56.5	90.4	59.2	38.2	57.8	47.7
+ ECBAM	10.25 M	2.92 G	57.3	90.0	63.3	38.2	58.7	48.0
TE-YOLOF-B3	16.76 M	6.6 G	58.4	90.6	66.0	40.4	59.6	47.5
- dilated encoder	13.28 M	3.12 G	57.6	90.2	61.7	34.4	61.0	49.2
+ BAM	13.28 M	3.12 G	57.2	90.3	62.8	39.0	59.1	46.7
+ CBAM	13.31 M	3.12 G	57.4	90.5	63.4	40.4	58.5	44.6
+ ECBAM	13.31 M	3.12 G	57.3	90.5	65.2	39.3	60.6	44.9

**Table 2 ijms-23-13355-t002:** Comparison with different Channel Attention Modules in TE-YOLOF without dilated encoder. The ‘*’ represents TE-YOLOF-B0 without dilated encoder and using 1×1 conv to reduce the channel number to default 512. Specifically, ‘+ ECA’ represents the Efficient Channel Attention used in [29].

Model	Params	GFLOPS	${\mathrm{AP}}_{50}$
TE-YOLOF *	6.46 M	2.73 G	89.7
+ ECA [29]	6.46 M	2.73 G	89.0
+ CAM in CBAM [17]	6.49 M	2.73 G	89.9
+ SE [16]	6.49 M	2.73 G	90.0
+ ECAM	6.49 M	2.73 G	90.1

**Table 3 ijms-23-13355-t003:** Add Spatial Attention Module or not. ‘TE-YOLOF *’ represents the same setting in TE-YOLOF-B0 used in Table 2. Each grid above is used to compare adding a spatial attention module or not in different channel attention mechanisms.

Model	Params	GFLOPS	${\mathrm{AP}}_{50}$
TE-YOLOF *	6.46 M	2.73 G	89.7
+ CAM in CBAM [17]	6.49 M	2.73 G	89.9
+ CBAM	6.5 M	2.73 G	90.3
+ SE [16]	6.49 M	2.73 G	90.0
+ SE + SAM in CBAM	6.5 M	2.73 G	90.2
+ ECAM	6.49 M	2.73 G	90.1
+ ECBAM	6.5 M	2.73 G	90.3

**Table 4 ijms-23-13355-t004:** Ablation study of spatial attention module kernel size in ECBAM. Kernel size represents the kernel size of convolution layer used in the spatial attention module in ECBAM.

Kernel Size	Params	GFLOPS	${\mathrm{AP}}_{50}$
3	6.49 M	2.73 G	89.8
5	6.49 M	2.73 G	90.2
7	6.5 M	2.73 G	90.3
9	6.5 M	2.73 G	90.2
11	6.5 M	2.73 G	89.7

**Table 5 ijms-23-13355-t005:** Experiments on the reduction ratio of MLP in ECAM.

Reduction Ratio	Params	GFLOPS	${\mathrm{AP}}_{50}$
2	6.72 M	2.73 G	89.8
4	6.59 M	2.73 G	89.3
8	6.53 M	2.73 G	90.0
16	6.5 M	2.73 G	90.3
32	6.48 M	2.73 G	90.0

## Data Availability

Not applicable.

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
