# Peer review of "Rethinking the Dilated Encoder in TE-YOLOF: An Approach Based on Attention Mechanism to Improve Performance for Blood Cell Detection"

_ijms, 2022, doi:10.3390/ijms232113355_

Round 1

Reviewer 1 Report

The paper by Xu et al., is an interesting reproposal of their original TE-YOLOF with attention mechanism. They first showed that the original dilated encoder in the module did not contribute to the performance. They then showed that the modified model called Enhanced Channel Attention Module (ECAM) can be more effective in addressing the blood detection problem. The paper gave excellent background introduction on the attention mechanism in CNN. Below are my comments:

1. Figure 1 should be better explained to audience and this comment also pertain to the section that describes this figure. There is no axis labels on both axis. I believe the x-axis are different backbone models. If that is the case, I would recommend using a barplot instead.

2. Several jargons/terms used in machine vision should be explained. Would be great if the authors can explain the metrics APs and the backbone model B.

3. They showed in Figure 1 that there is no difference in performance using dilated encoder but did not specified what datasets were evaluated and if the result is dataset dependent.

4. Also, what are the size of the datasets and how are the images splitted? What are the proportion of training v. validation v. test set from BCCD?

5. As described in the legend of Table 1, the model were evaluated using the validation set. How about the test set performance? Also are the results in Table 2-4 using the validation set as well?

6. There are many grammar errors in the text, I would suggest more a careful revision.

Minor comments:

1. line 176, “a question that naturally raised is whether”

2. line 177, “effective for performance improvement

3. line 171, “we turn our attention to the…”

Reviewer 2 Report

The authors wanted to investigate the effect and benefits of removing dilated encoder module in TEYOLOF architecture for blood cell detection. I think the paper is well written, the findings are presented succinctly, and it is ready for publication.

One observation: In fig.1 the Y-axis is missing a title and the range should be lowered so that readers are able to see the difference between the two variations
